# When low-vision task meets dense prediction tasks with less data: an auxiliary self-trained geometry regularization

**Zaiwang Gu**    *Gu_Zaiwang@i2r.a-star.edu.sg*
*Institute for Infocomm Research ($I^2R$), Agency for Science, Technology and Research (A\*STAR), Singapore*

**Weide Liu**    *weide.liu@childrens.harvard.edu*
*Boston Children's Hospital and Harvard Medical School, Boston, MA*

**Xulei Yang**    *yang_xulei@i2r.a-star.edu.sg*
*Institute for Infocomm Research ($I^2R$), Agency for Science, Technology and Research (A\*STAR), Singapore*

**Chuan-Sheng Foo**    *foo_chuan_sheng@i2r.a-star.edu.sg*
*Institute for Infocomm Research ($I^2R$), Agency for Science, Technology and Research (A\*STAR), Singapore*
*Centre for Frontier AI Research (CFAR), Agency for Science, Technology and Research (A\*STAR), Singapore*

**Jun Cheng** [*]    *Cheng_Jun@i2r.a-star.edu.sg*
*Institute for Infocomm Research ($I^2R$), Agency for Science, Technology and Research (A\*STAR), Singapore*

**Reviewed on OpenReview:** *https://openreview.net/forum?id=44qpZ6pkau*

## Abstract

Many deep learning methods are data-driven, often converging to local minima due to limited training data. This situation poses a challenge in domains where acquiring adequate data is difficult for model training or fine-tuning, such as generalized few-shot semantic segmentation (GFSSeg) and monocular depth estimation (MDE). To this end, we propose a self-trained geometry regularization framework to enhance model training or fine-tuning in scenarios with limited training data using geometric knowledge. Specifically, we propose to leverage low-level geometry information extracted from the training data and define a novel regularization term, which is a plug-and-play module jointly trained with the primary task via multi-task learning. Our proposed regularization neither relies on extra manual labels and data in training nor requires extra computation during the inference stage. We demonstrate the effectiveness of this regularization on GFSSeg and MDE tasks. Notably, it improves the state-of-the-art GFSSeg by 5.61% and 4.26% mIoU of novel classes on PASCAL and COCO in the 1-shot scenario. In MDE, it achieves a relative reduction of SILog error by 16.6% and 9.4% for two recent methods in the KITTI dataset.

## 1 Introduction

The recent advancements in deep learning-based approaches have been significant and promising. Numerous modern and sophisticated deep neural networks (Dosovitskiy et al., 2020; Liu et al., 2021; Oquab et al., 2023; Carion et al., 2020; Touvron et al., 2022) have been proposed to address a variety of tasks. These deep learning models are mostly data-driven and require large amounts of training data (Zhou et al., 2017; Deng et al., 2009; Russakovsky et al., 2015; Kuznetsova et al., 2020), to avoid the risk of converging to local minima. However, annotating large quantities of training samples in practice is exceedingly challenging, particularly for tasks involving dense predictions. For instance, in monocular depth estimation (MDE), collecting large-scale data with paired ground truth is costly (Ranftl et al., 2022; Bhat et al., 2023; Ranftl et al., 2021a; Piccinelli et al., 2024). Additionally, in generalized few-shot semantic segmentation (GFSSeg) (Tian et al.,

---

[*]corresponding author

2022; Hajimiri et al., 2023; Liu et al., 2023a; Lang et al., 2022), only a few shots of data are available to adapt the models to novel classes. As a result, significant challenges remain in dense prediction tasks with less data.

Training with less data often leads to local minima convergence. To address this issue, one way is to train the foundation models to learn task-invariant features via self-supervised learning on large-scale unlabeled data (Chen et al., 2020; Caron et al., 2021; Bao et al., 2022; He et al., 2022). These models can then be fine-tuned on smaller, task-specific annotated datasets for downstream tasks. In (Darcet et al., 2023; Oquab et al., 2023), robust visual features are first learned without supervision, followed by evaluations on two dense downstream tasks: semantic image segmentation and depth estimation.

Another approach to mitigate local minima convergence is to employ geometry regularization. Encoding geometry constraints through auxiliary learning enables the model to capture the task-invariant features, thereby improving the convergence of the primary task. Edge or semantic information can be considered as examples of geometry information. Several earlier works (Ramirez et al., 2018; Fu et al., 2018b; Zhang et al., 2019a; Zhu et al., 2020) have demonstrated that jointly detecting edge and semantic labels enhances performance. Previous researchers have used edge as a supervisor signal to improve segmentation (Yu et al., 2018) and depth estimation (Zhu et al., 2020). Other methods leverage edges or semantics to guide the primary tasks (e.g., depth estimation) through iterative fine-tuning (Shao et al., 2023b; Xu et al., 2023a) or other post-processing techniques (Su & Tao, 2023).

However, existing foundation model-based approaches require carefully designed regularization modules to prevent representation collapse (Grill et al., 2020; Caron et al., 2021; He et al., 2020; Oquab et al., 2023) and may also be prone to catastrophic forgetting in model fine-tuning (Li et al., 2023b; Xu et al., 2023b). While geometry regularization is generally effective, the difficulty lies in acquiring different geometry constraints. Semantic labels often provide more geometric information than edges, but they are also more challenging to acquire. Previous approaches (Ramirez et al., 2018; Fu et al., 2018b; Zhang et al., 2019a; Zhu et al., 2020) often incorporate low-level geometry information into the primary task for feature fusion, which raises the cost of computation during both training and inference stages.

We propose a novel auxiliary plug-and-play geometry regularization network (GRNet) to enhance model training. GRNet is discardable and does not incur additional computational costs during inference. In addition, we propose a novel primary-to-auxiliary aggregation module within GRNet to account for the relationship between the primary task and the regularization task. By rethinking geometry constraints, we aim to introduce an accessible yet effective low-level geometry regularization. We explore different low-level geometry information, while we hypothesise that the edge is just a special case. Specifically, in the auxiliary regularization term, we investigate keypoints, e.g., SuperPoint (DeTone et al., 2018), scare-invariant feature transform (SIFT) (Lowe, 2004), and edges, e.g., Canny edges (Canny, 1986), as pseudo ground truths.

Based on recent progress in model-based deep learning (Shlezinger et al., 2020), additional regularization often enhances the model training and reduces overfitting. The fact that regularization improves performance validates that the deep learning model trained on small datasets can be biased. Our research indicates that model bias is another major factor compromising performance. Including additional terms, such as edge, is a special case that would improve model generalization and, therefore, better performance. To further validate our hypothesis, we utilize keypoints, which are not directly related to segmentation, and demonstrate that they also contribute to improved results.

The proposed GRNet offers three key benefits. Firstly, it enhances the feature extraction in the backbone, making it more effective for multiple tasks. Secondly, by integrating the output of the primary task into an auxiliary low-level structure detection task, we can impose a constraint on the output of the primary task such that it favours low-level geometry extraction in the auxiliary task. This auxiliary geometry aggregation module would also be discarded after the training, ensuring no additional computational cost during inference. A minor limitation of the proposed approach is the increase in training time. Interestingly, different choices of geometry knowledge, such as edges and key points, comparably improve the performance of the main tasks. This finding offers new insights into the role of auxiliary edge detection in previous methods, which we will discuss in more detail in Section 4.1.4 and Section 4.2.3. Thirdly, obtaining low-level structures for regularization is easier than acquiring semantic labels. For instance, the edges can be

estimated by pretrained deep learning models or traditional handcrafted methods such as the Canny edge detector (Canny, 1986). The key points can be easily computed by SIFT (Lowe, 2004) and self-supervised SuperPoint model (DeTone et al., 2018).

The major contributions of this paper are:

- We propose a simple yet effective self-trained geometry regularization, which uses the accessible low-level geometry information to construct a regularization term via multi-task learning to train dense prediction tasks with less data.

- We propose a primary-to-auxiliary geometry aggregation module to account for the relationship between the primary and auxiliary tasks. This design encourages the primary branch to generate predictions that benefit the auxiliary task, which differs from post-processing fusion or feature smoothing in previous methods.

- We integrate the proposed regularization term with various approaches in GFSSeg and MDE tasks. Experimental results demonstrate that it is generic for different architectures and tasks.

## 2 Related works

### 2.1 Dense Prediction Tasks with Less Data

**Dense prediction tasks** refer to tasks where predictions are required for each pixel, such as depth estimation (Laina et al., 2016; Yuan et al., 2022; Patil et al., 2022; Shao et al., 2023a; Yang et al., 2024; Ranftl et al., 2020; Bhat et al., 2023; Li et al., 2023a; Kirillov et al., 2023), semantic image segmentation (Zhao et al., 2017; Chen et al., 2017; Cheng et al., 2022; Xie et al., 2021; Li et al., 2023a; Isensee et al., 2021), etc. Recently, The Segment Anything model and Depth Anything model have shown that large-scale data is important in improving the performance of dense prediction tasks (Ranftl et al., 2021b; Kirillov et al., 2023; Bhat et al., 2023; Yang et al., 2024). However, acquiring large-scale datasets for certain tasks can be challenging. For instance, around 63 million images are collected for training the Depth Anything model (Yang et al., 2024), and 12 million for the Segment Anything model (Kirillov et al., 2023). Training with such large-scale data is resource-intensive. Instead of pursuing large-scale data, we aim to design accessible yet effective geometry regularization for dense prediction tasks with less data. In this paper, we use generalized few-shot semantic segmentation (GFSSeg) and monocular depth estimation (MDE) as two case studies to validate our approach, where both tasks face data scarcity issues.

**Generalized few-shot semantic segmentation** is a type of dense prediction task with less data (Tian et al., 2022; Myers-Dean et al., 2021; Liu et al., 2023a; 2024; 2022). These methods adapt models to novel classes while maintaining performance on base classes, using only a few shots of data from the novel classes. Tian et al. (2022) propose context-aware prototype learning (CAPL), which mines contextual cues from support and query samples to enrich the classifier for the novel class segmentation. Liu et al. (2023a) introduce projection onto orthogonal prototypes (POP) to generalize well on base classes and quickly adapt to new objects or instances. Hajimiri et al. (2023) propose leveraging the InfoMax principle to maximize Mutual Information (MI) between learned feature representations and predictions with an easily optimized inference phase. We utilize two methods (CAPL and POP) as baselines to assess the effectiveness of our method in GFSSeg.

**Monocular depth estimation** is another example. Some approaches (Ranftl et al., 2020; Bhat et al., 2023; Yang et al., 2024; Piccinelli et al., 2024) aim to collect as much data as possible to train robust models. Ranftl et al. (2020) attempt to combine multiple datasets from different sources and propose a robust training objective to boost the depth estimation performance. Bhat et al. (2023) propose the first approach that combines both relative and metric depth estimations, resulting in a model with excellent generalization performance in MED while maintaining metric scale. Built upon (Bhat et al., 2023), Yang et al. (2024) extended the pretraining dataset to 62 million images. This model is then fine-tuned using a smaller dataset of NYUv2 or KITTI, achieving remarkable generalization performance. Since the data for

model fine-tuning is much smaller than the data used for pretraining, we consider this a low-data scenario. Using such large-scale datasets introduces significant challenges for data collection and model training.

Recent other works (Shao et al., 2023b; Yuan et al., 2022; Shao et al., 2023a) have increasingly employed the vision transformer. Leveraging the pretrained transformer (Liu et al., 2021) as a backbone encoder, Yuan et al. (2022) proposed neural window fully-connected conditional random fields (NeWCRFs) as depth decoder, showing promising performance in MDE task. Building on NeWCRFs, Shao et al. (2023a) introduce a physics-based regularization method incorporating a normal distance map for depth estimation. Shao et al. (2023b) propose Iterative elastic Bins (IEBins), which reformulates the regression task into the classification task and adopts an iterative post-processing to improve the results. However, it requires 129% extra computational cost. In this work, we use NeWCRFs as our baseline, and we find that our GRNet can improve the NeWCRFs by 7.4% and 16.6% in NYUv2 (Nathan Silberman & Fergus, 2012) and KITTI (Geiger et al., 2013). Compared with IEBins, GRNet with NeWCRFs can achieve comparable or superior performance without using its iterative post-processing and, therefore, avoids the 129% additional computational cost during inference. A limitation is that our proposed approach increases training time, although it does not affect computational requirements during inference.

## 2.2 Auxiliary Edge and Semantic Detection

**Semantic information** has been utilized to improve the performance of main tasks. Ramirez et al. (2018) propose to leverage the semantics and geometry by enforcing spatial proximity between depth discontinuities and semantics for monocular depth estimation. Wu et al. (2019) introduce pyramid cost volumes to capture semantic and multi-scale spatial information for semantic stereo-matching. Zhu et al. (2020) explore the constraints from semantic segmentation to help unsupervised monocular depth estimation. Rahman & Fattah (2024) propose a depth semantics symbiosis module to achieve comprehensive mutual awareness information for boosting depth estimation. While semantic labels are beneficial, acquiring large-scale labels can be costly in many cases.

**Low-level structure information** is more accessible to compute compared to semantics. For instance, it is easy to compute the Canny edge or Sobel edge using the famous Canny operator (Canny, 1986) and the Sobel operator (Sobel & Feldman, 1973). Pretrained deep-learning models, such as SAM (Kirillov et al., 2023), can also be employed. Schenk & Fraundorfer (2017) leverage the combination of edge images and depth maps for joint camera pose estimation. In some recent work, synthetic images are used to train deep learning models, where the ground truths can be easily generated (Zhang et al., 2019b; DeTone et al., 2018). In this work, we explore using different low-level geometric information, such as keypoints and the edge, in the auxiliary task. The previously used edge is considered a special case. Our study demonstrates that diverse low-level geometries work similarly in improving the models.

## 3 Geometry Regularization Network

In Section 3.1, we first describe our proposed geometry regularization network (GRNet) and its optimization objective. Next, we elaborate on the design of the geometry regularization module in Sections 3.2 and 3.3. Finally, we discuss the choice of low-level geometry information in Section 3.4.

## 3.1 The Overall Architecture and Optimization Objective

We propose a simple yet effective geometry regularization network (GRNet) to improve the model training or fine-tuning in dense prediction tasks with limited training data, specifically in generalized few-shot semantic segmentation (GFSSeg) and monocular depth estimation (MDE) tasks. GRNet constructs a regularization term by learning from low-level keypoints or edge information via multi-task learning, as illustrated in Figure 1. We denote the primary dense prediction task as the primary task and the regularization term as the auxiliary task. Both tasks are trained jointly using multi-task learning.

Our optimization objective is to use the auxiliary task to regularize the primary task, combining the losses of both tasks through a weighted sum. Supervisory geometric information is obtained from hand-crafted or

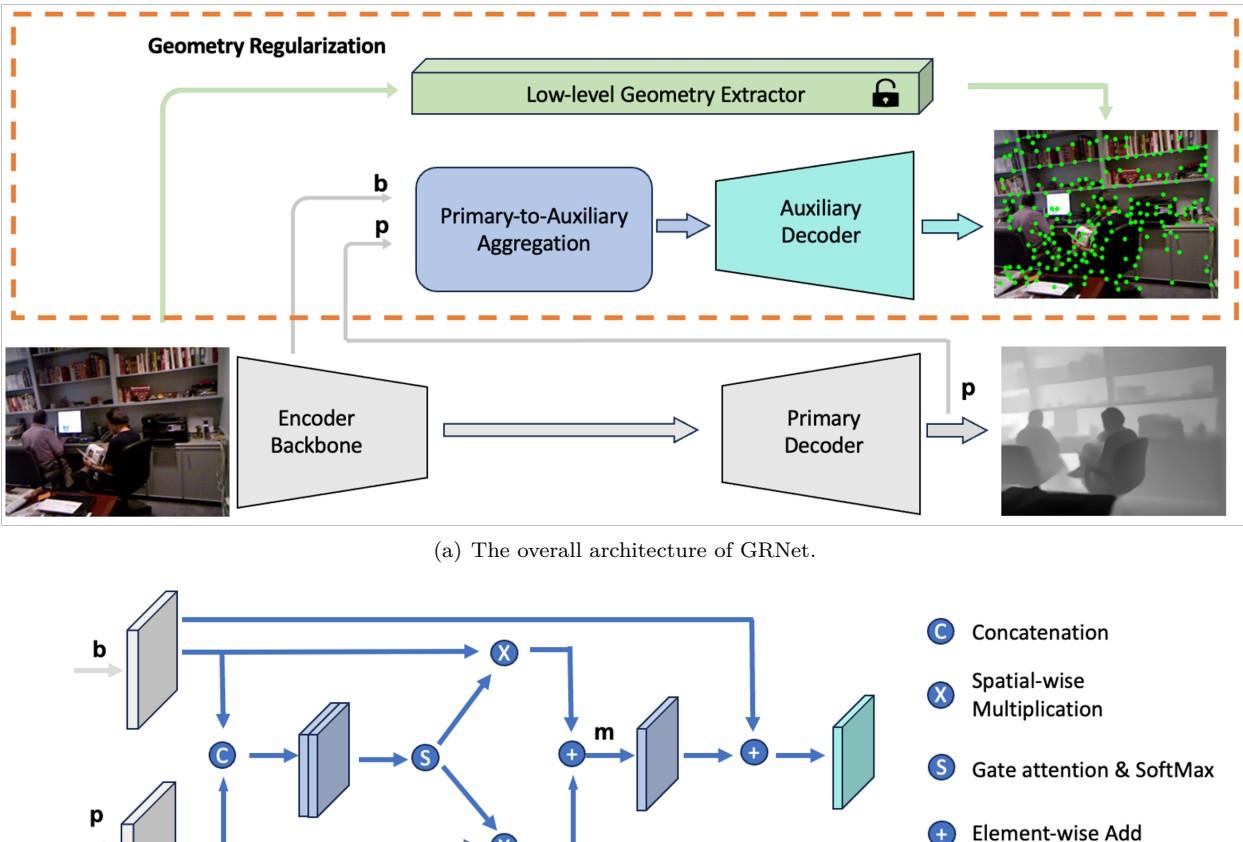

(a) The overall architecture of GRNet.

(b) The details of primary-to-auxiliary aggregation.

Figure 1: The pipeline of the proposed auxiliary self-trained geometry regularization network (GRNet). (a) GRNet consists of a low-level geometry extractor, a primary-to-auxiliary aggregation module, and an auxiliary decoder. (b) The primary-to-auxiliary aggregation module first combines two inputs, feature **b** from the encoder backbone and output **p** of the primary task, followed by gate attention and softmax to generate the correlation attention maps between primary and auxiliary tasks.

pretrained low-level extractors. We adopt the focal loss (Lin et al., 2020) to compute the loss $L_{gr}$ for the geometry regularization, as shown in Equation (1):

$$L_{gr} = \mathcal{F}(s_{gt}, s_{est}), \tag{1}$$

where $s_{gt}$ and $s_{est}$ denote the pseudo low-level geometry supervision and estimated geometry prediction, respectively. $\mathcal{F}$ denotes the function to compute focal loss.

The overall loss is computed as the weighted sum of the loss $L_p$ from the primary dense prediction decoder and $L_{gr}$ from GRNet, as shown in Equation (2):

$$L = L_p + \lambda \cdot L_{gr}, \tag{2}$$

where $\lambda$ controls the balance of the two items and $\lambda$ is set to 1 empirically in our work.

## 3.2 The Auxiliary Geometry Regularization Structure

The proposed geometry regularization module includes a low-level geometry extractor, a primary-to-auxiliary aggregation module and an auxiliary decoder, as shown within the dashed orange line in Figure 1(a). In this

work, the low-level geometry extractor is implemented with a self-supervised model or hand-crafted methods to obtain the keypoints or edge information. This low-level information is treated as pseudo-ground truth, requiring no manual labeling. The low-level geometry extractor is a non-trainable module, and there are several options, such as SIFT (Lowe, 2004), SuperPoint (DeTone et al., 2018) and Canny (Canny, 1986), which we will further discuss in Section 3.4. The primary-to-auxiliary aggregation module takes the output of the primary task as one of the inputs of the auxiliary task. It accounts for the relationship between low-level structures and the primary task, which will be elaborated on in Section 3.3. The auxiliary decoder follows a simple design consisting of three decoder blocks. Each block includes a $1 \times 1$ convolution, a $3 \times 3$ transposed convolution and a $1 \times 1$ convolution consecutively.

### 3.3 Primary-to-Auxiliary Aggregation

Unlike other methods that fuse the outputs of the low-level tasks for the primary dense prediction task, we propose a regularization approach for the primary dense prediction task, as shown in Figure 1(a). We propose a primary-to-auxiliary (P2A) aggregation module with soft attention to achieve a one-way information flow from the primary dense prediction branch to the auxiliary low-level regularization branch. Figure 1(b) illustrates the detailed diagram of the proposed P2A aggregation module. Given feature $\mathbf{b}$ extracted from the backbone and output $\mathbf{p}$ from the primary decoder, we first concatenate them to obtain $\mathbf{c} = Concat(\mathbf{b}, \mathbf{p})$. Then we define two functions to map $\mathbf{c}$ via two different spatial-wise gates $G_{\mathbf{b}}$ and $G_{\mathbf{p}}$ via two $1 \times 1$ convolution. This results in:

$$A_{\mathbf{b}} = G_{\mathbf{b}}(\mathbf{c}), A_{\mathbf{p}} = G_{\mathbf{p}}(\mathbf{c}). \tag{3}$$

Next, the softmax function is applied to $A_{\mathbf{b}}$ and $A_{\mathbf{p}}$ to generate $S_{\mathbf{b}}$ and $S_{\mathbf{p}}$.

$$S_{\mathbf{b}} = \frac{e^{A_{\mathbf{b}}}}{e^{A_{\mathbf{b}}} + e^{A_{\mathbf{p}}}}, S_{\mathbf{p}} = \frac{e^{A_{\mathbf{p}}}}{e^{A_{\mathbf{b}}} + e^{A_{\mathbf{p}}}}. \tag{4}$$

A merged feature $\mathbf{m}$ can be computed by a weighted sum of the feature $\mathbf{b}$ and output $\mathbf{p}$:

$$\mathbf{m} = \mathbf{b} \cdot S_{\mathbf{b}} + \mathbf{p} \cdot S_{\mathbf{p}}. \tag{5}$$

We then calculate $\mathbf{b}'$ as average of $\mathbf{m}$ and the feature $\mathbf{b}$ for subsequent low-level structure detection.

$$\mathbf{b}' = \frac{\mathbf{b} + \mathbf{m}}{2}. \tag{6}$$

This module is introduced to regularize the model training and prevent convergence to local minima. It seamlessly integrates the existing methods in few-shot segmentation and monocular depth estimation without changing the network structure of existing methods. As a result, this plug-and-play module is discarded after training, incurring no additional computational cost during inference.

It is important to emphasize that our method is conceptually distinct from many existing methods (Li et al., 2020; Lu et al., 2022; Li et al., 2024). Our method utilizes the output $\mathbf{p}$ of the primary task as one of the inputs of the auxiliary task. This design encourages the network to compute $\mathbf{p}$ in a way that benefits the auxiliary task, i.e., the network would compute $\mathbf{p}$ that leads to lower loss of the auxiliary task. This is equivalent to imposing a constraint on the output $\mathbf{p}$ such that it favours low-level geometry extraction in the auxiliary task. In contrast, other methods often apply post-processing smoothing directly to the output.

### 3.4 Choice of Low-level Geometry Information

One important aspect of the method is the type of geometry information used to train the auxiliary decoder. Our study explores several options, including SIFT, SuperPoint and Canny edge. Previous studies (Song et al., 2020) often conjecture that joint edge detection provides information to smooth the output of the primary task and improves the results. Some other methods (Shao et al., 2023b) use such information to iteratively update MDE. In GRNet, we find that the keypoints and edge work similarly to improve the results in GFSS and MDE tasks. This raises the question of the role of edge detection in previous methods. We have conducted some experiments in our study in Section 4.2.3, and our experiments suggest that post-processing iterations could be eliminated if the proposed GRNet is used to regularize model training.

## 4 Experimental Results

We evaluate GRNet on generalized few-shot semantic segmentation (GFSSeg) and monocular depth estimation (MDE). We release our code at github [1].

### 4.1 Generalized Few-shot Semantic Segmentation

#### 4.1.1 Datasets and Evaluations

- **Datasets.** We use the PASCAL-$5^i$ dataset, built on the PASCAL VOC 2012 (Everingham et al., 2010). This dataset contains 12,031 images with high-quality pixel-level annotations of 20 classes, split into 10,582 training images and 1,449 validation images. Following the standard protocol in (Liu et al., 2023b; Tian et al., 2022), the 20 classes are evenly partitioned into four folds for cross-validation. Additionally, we evaluate the capability of our method on a more challenging COCO-$20^i$ dataset, which includes 122,218 images with 80 object classes, with 82,081 images used for training and 40,137 for validation. Similarly, experiments on COCO-$20^i$ are also conducted with cross-validation on four folds (20 classes per fold).

- **Evaluation Metrics.** For both datasets, once the model is validated on a given fold, the classes in this fold are treated as novel classes, and the classes in the other three folds plus background play the role of base classes. The Intersection over Union (IoU) per class and mean IoU (mIoU) over the base, novel, and all classes are computed. We follow the same settings as in (Tian et al., 2022; Myers-Dean et al., 2021; Liu et al., 2023a) to conduct four-fold cross-validation and calculate the average values. Additionally, each experiment is repeated with five different random seeds, and the mean results are reported (Tian et al., 2022).

#### 4.1.2 Implementation Details

Our GRNet is implemented on PyTorch. Following the training strategy in (Tian et al., 2022; Liu et al., 2023a), the mini-batch stochastic gradient descent with momentum 0.9 and weight decay 0.0001 is used to optimize the model. During the base class learning, the initial learning rate is set to 0.01, which is annealed down to zero following a "poly" policy whose power is fixed to 0.9. The batch size is set to 8 for both datasets, and the models are trained for 100 epochs for base class learning. For novel class updating, the model is updated with a fixed learning rate of 0.01 for 500 epochs, and the batch size is set as 2 and 8 for PASCAL$5^i$ and COCO-$20^i$, respectively.

#### 4.1.3 Comparison with state-of-the-art methods

We integrate the proposed GRNet with the two latest GFSSeg methods, CAPL (Tian et al., 2022) and POP (Liu et al., 2023a), to justify its effectiveness on PASCAL-$5^i$ and COCO-$20^i$ datasets. We denote the method applying GRNet on baseline approach A as "GRNet w A". We choose the SIFT keypoint as the main geometry knowledge in our implementation, but the results using other low-level structures are also reported. Table 1 summarizes the mIoU performances of different settings ($k$-shot, $k = 1, 5, 10$) on the PASCAL-$5^i$ dataset. Overall, the results across three different settings consistently indicate that the proposed GRNet can improve the performances of the novel classes. In particular, the mIoU performance of novel class by GRNet with the CAPL outperforms the original CAPL model by 7.1%, 27.5% and 14.1% relatively in 1-shot, 5-shot and 10-shot, respectively, while the performances for base classes are maintained or slightly improved. Similarly, the GRNet with POP outperforms the original POP by 15.8%, 2.2%, and 1.8% for novel classes in 1-shot, 5-shot, and 10-shot, respectively.

Table 1 shows that GRNet with CAPL outperforms the baseline CAPL method on base and novel classes across three different shot experiments. In contrast, the results of GRNet with POP show only a slight mIoU improvement (0.02%) of base class on the 5-shot setting, and the performance is even lower in the 1-shot and 10-shot settings. We think the reason is the different training strategies between CAPL and POP. Notably,

---

[1] https://github.com/Guzaiwang/Geometry_Regularization

Table 1: Performance comparisons on PASCAL-$5^i$ for GFSSeg. We report mIoU (%) over base classes (Base), novel classes (Novel), and all classes (Base + Novel = Total). All models are based on ResNet-50. † represents the use of SIFT as the constraint of our GRNet. "GRNet w A" denotes to apply GRNet on A.

| Method | 1-shot | | | 5-shot | | | 10-shot | | |
|---|---|---|---|---|---|---|---|---|---|
| | Base | Novel | Total | Base | Novel | Total | Base | Novel | Total |
| PFENet (Tian et al., 2020) | 8.32 | 2.67 | 6.97 | 8.83 | 1.89 | 7.18 | / | / | / |
| PANet (Wang et al., 2019) | 31.88 | 11.25 | 26.97 | 32.95 | 15.25 | 28.74 | / | / | / |
| DIaM (Hajimiri et al., 2023) | 70.89 | 35.11 | 61.24 | 70.85 | 55.31 | 68.29 | / | / | / |
| FT (Myers-Dean et al., 2021) | 66.84 | 18.82 | 55.41 | 72.03 | 46.40 | 65.93 | 73.02 | 52.55 | 68.14 |
| FT-Triplet | 66.41 | 19.71 | 55.31 | 71.31 | 50.46 | 66.35 | 72.87 | 57.00 | 69.10 |
| CAPL (Tian et al., 2022) | 65.48 | 18.85 | 54.38 | 66.14 | 22.41 | 55.72 | 69.09 | 27.17 | 59.11 |
| GRNet w CAPL † | **69.72** | **20.18** | **57.92** | **70.51** | **28.57** | **60.53** | **72.51** | **31.01** | **62.82** |
| POP (Liu et al., 2023a) | **73.92** | 35.51 | 64.77 | 74.78 | 55.87 | 70.28 | **74.99** | 58.77 | 71.13 |
| GRNet w POP † | 73.75 | **41.12** | **66.01** | **74.80** | **57.12** | **70.89** | 74.83 | **59.81** | **71.52** |

Table 2: Performance comparisons on COCO-$20^i$ for GFSSeg. Models are based on ResNet-50. † represents the use of SIFT as the constraint of our GRNet.

| Method | 1-shot | | | 5-shot | | | 10-shot | | |
|---|---|---|---|---|---|---|---|---|---|
| | Base | Novel | Total | Base | Novel | Total | Base | Novel | Total |
| DIaM | 48.28 | 17.22 | 40.29 | 48.37 | 28.73 | 46.72 | / | / | / |
| FT | 43.42 | 8.94 | 34.90 | 47.18 | 24.72 | 41.63 | 48.18 | 30.03 | 43.70 |
| FT-Triplet | 43.64 | 9.23 | 35.14 | 46.61 | 28.84 | 41.36 | 46.61 | 34.49 | 43.27 |
| CAPL (Tian et al., 2022) | 44.61 | 7.05 | 35.46 | 45.24 | 11.05 | 36.80 | 45.51 | 10.82 | 36.95 |
| GRNet w CAPL † | **46.26** | **9.36** | **37.16** | **46.38** | **13.30** | **38.21** | **46.38** | **14.55** | **39.63** |
| POP (Liu et al., 2023a) | **54.71** | 15.31 | 44.98 | **54.90** | 29.97 | 48.75 | 55.01 | 35.05 | 50.08 |
| GRNet w POP † | 53.81 | **19.57** | **45.36** | 54.12 | **31.62** | **49.27** | **55.06** | **36.18** | **50.29** |

GRNet causes a slight performance drop for base classes in POP (Liu et al., 2023a), but this does not happen in CAPL (Tian et al., 2022). This is due to the different training strategies between CAPL and POP. CAPL iteratively optimizes the models for the base classes and updates the novel classes in each epoch. POP trains the models for base classes first and then updates them for the novel classes without utilizing the base class data during the update.

Table 2 details the performance comparisons of 1-shot, 5-shot, and 10-shot on COCO-$20^i$ dataset. Similar to the experimental results on the PASCAL-$5^i$ dataset, GRNet with POP surpasses the original POP by 27.8%, 5.5%, and 3.2% relatively on 1-shot, 5-shot, and 10-shot settings. The results again empirically verify the effectiveness of our method. Figure 2 shows three sample results for GFSSeg in PASCAL, by CAPL, POP and our methods under the 5-shot setting. As shown in the results, our methods achieve more accurate segmentation.

Next, we delve into the deeper insights revealed by the experimental results. When training data is limited or scarce in the few-shot segmentation task, the risk of overfitting increases as the model tends to optimize for the limited training data, leading to poor generalization. While data augmentation can somewhat alleviate this issue, there is still space for improvement. Our core idea is to introduce an extra branch as the low-level geometry regularization. This branch is included via multi-task learning. By including the low-vision task in the few-shot semantic segmentation, we ask the network to optimize for both the primary and low-vision tasks simultaneously. Table 1 and 2 present that our proposed GRNet algorithm enhances the performance of novel classes across three different settings, demonstrating its effectiveness in reducing over-fitting and improving generalization. As the training data for novel classes increases, the gain provided by GRNet

Table 3: Performance comparisons of different geometry regularization on PASCAL-$5^i$, including SIFT, SuperPoint and Canny edge. We report mIoU (%) over base classes (Base), novel classes (Novel), and all classes (Base + Novel = Total). All models are based on ResNet-50. †, ‡ and § respectively represent the use of SIFT, SuperPoint and Canny edge as the constraints of our GRNet.

| Method | 1-shot | | | 5-shot | | | 10-shot | | |
|---|---|---|---|---|---|---|---|---|---|
| | Base | Novel | Total | Base | Novel | Total | Base | Novel | Total |
| CAPL (Tian et al., 2022) | 65.48 | 18.85 | 54.38 | 66.14 | 22.41 | 55.72 | 69.09 | 27.17 | 59.11 |
| GRNet w CAPL † | 69.72 | **20.18** | 57.92 | **70.51** | **28.57** | 60.53 | **72.51** | 31.01 | 62.82 |
| GRNet w CAPL ‡ | **70.12** | 20.03 | **58.19** | 70.47 | 29.49 | **60.72** | 72.48 | **31.25** | **62.96** |
| GRNet w CAPL § | 69.65 | 20.13 | 57.85 | 70.08 | 29.53 | 60.43 | 72.33 | 30.62 | 62.53 |
| POP (Liu et al., 2023a) | **73.92** | 35.51 | 64.77 | 74.78 | 55.87 | 70.28 | **74.99** | 58.77 | 71.13 |
| GRNet w POP † | 73.75 | **41.12** | 66.01 | 74.70 | **57.12** | **70.89** | 74.83 | **59.81** | **71.52** |
| GRNet w POP ‡ | 73.71 | 41.09 | **65.98** | 74.62 | 57.08 | 70.81 | 74.76 | 59.68 | 71.44 |
| GRNet w POP § | 73.83 | 40.52 | 65.89 | **74.88** | 56.99 | 70.65 | 74.98 | 59.67 | 71.34 |

reduces. Specifically, GRNet with POP improves the mIoU across all classes by 1.24%, 0.61%, and 0.38% in the 1-shot, 5-shot, and 10-shot settings, respectively.

### 4.1.4 Choice of low-level geometry extractor

Table 4: $p$ values of statistic t-test between SIFT and other regularizations in 1-shot CAPL on PASCAL dataset.

| SIFT vs. | Baseline | SuperPoint | Canny |
|---|---|---|---|
| $p$-value | 0.032 | 0.109 | 0.622 |

To verify the effectiveness of different low-level geometry extractors, we compare the SIFT (†), SuperPoint (‡) and Canny edge (§) as low-level geometry in the GFSSeg. Table 3 presents the performance comparisons of different geometry regularizations on PASCAL-$5^i$. In the GRNet with CAPL, the three different low-level geometries improve the performance of novel classes comparably. We conduct a statistic t-test to compare SIFT regularization against SuperPoint, Canny and the baseline without GRNet.

Table 4 shows the statistic t-test $p$ values to validate the choice of regularizations. As we can see, there is a significant difference between the SIFT regularized model and the baseline ($p < 0.05$). The different choices of regularization terms do not lead to significant differences ($p > 0.05$), suggesting that the choice does not critically impact the final performance.

Table 5: Ablation study for GRNet on PASCAL in the generalized few-shot segmentation.

| Baseline (POP) | Multi-task | Primary-to-Auxiliary | 1-shot Base/Novel/Total | 5-shot Base/Novel/Total | 10-shot Base/Novel/Total |
|---|---|---|---|---|---|
| ✓ | | | **73.92**/35.51/64.77 | 74.78/55.87/70.28 | **74.99**/58.77/71.13 |
| ✓ | ✓ | | 73.63/36.88/64.89 | 74.69/56.47/70.36 | 74.89/59.24/71.27 |
| ✓ | ✓ | ✓ | 73.75/**41.12/66.01** | **74.70/57.12/70.89** | 74.83/**59.81/71.52** |

### 4.1.5 Ablation Studies

We conduct ablation studies to evaluate the effectiveness of each proposed module. We use POP as the baseline. The GRNet includes two modules. The first module is a simple multi-task learning of the primary task and a low-level extraction task to detect SIFT, denoted as "Multi-task". The second module is the primary-to-auxiliary module, which accounts for the relationship between the two tasks. Table 5 summarizes the performance comparison when the two components are applied in GFSSeg. Specifically, in the case of

5-shot segmentation, adding the "Multi-task" component increases the IoU metric of novel classes from 55.87 to 56.47. Furthermore, integrating the primary-to-auxiliary module results in a further improvement, raising the final IoU of novel classes to 57.12. The results show that both components improve the performance of novel classes without significantly affecting the accuracy of the base classes. Multi-task learning and the primary-to-auxiliary aggregation module enhance the model's generalization ability.

Next, we investigate the impact of different $\lambda$ weights on the results. Table 6 presents the performance comparisons for different hyperparameters $\lambda$ choices. The results indicate that when $\lambda$ is set to 1, the proposed algorithm achieves the mIoU of 70.87 in 5-shot segmentation on the PASCAL dataset, outperforming the settings where $\lambda$ is 0.1 and 0.5.

Table 6: Performance comparisons of different hyper-parameters on PASCAL-$5^i$ with SIFT. We report mIoU (%) of all classes (Base + Novel = Total)

| $\lambda$ | 0.1 | 0.5 | 1 |
|---|---|---|---|
| GRNet w POP † | 70.77 | 70.68 | 70.89 |

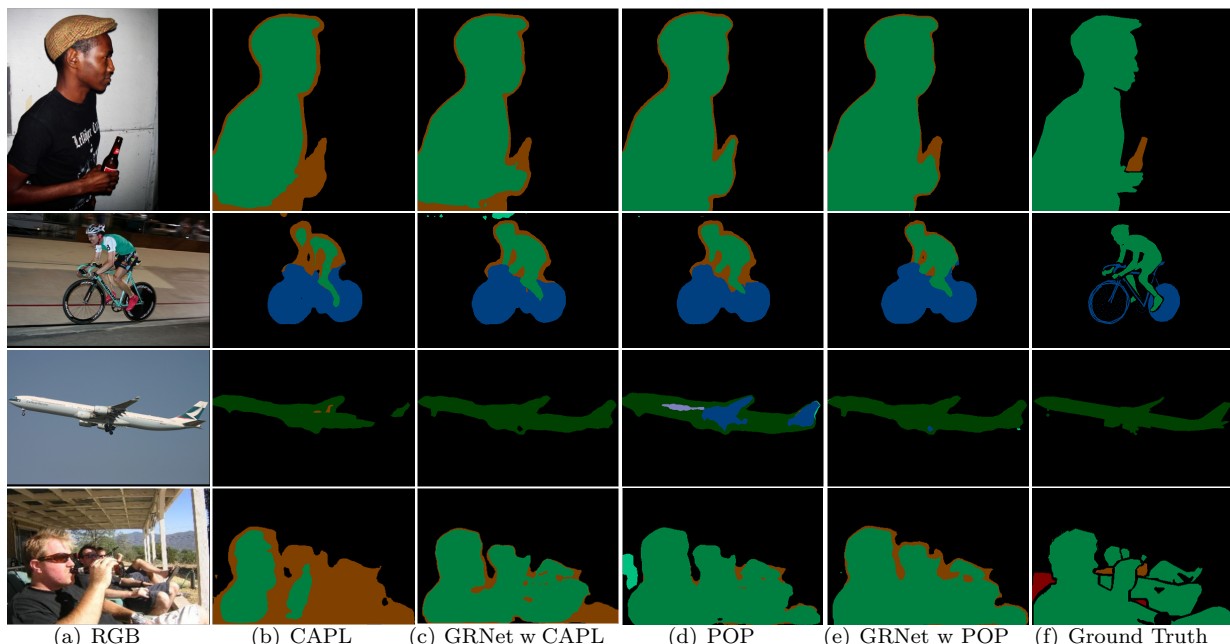

(a) RGB  (b) CAPL  (c) GRNet w CAPL  (d) POP  (e) GRNet w POP  (f) Ground Truth

Figure 2: The visualization comparison of three examples from PASCAL by different methods under the 5-shot setting of GFSSeg.

## 4.2 Monocular Depth Estimation

### 4.2.1 Datasets

We use the NYUv2 (Nathan Silberman & Fergus, 2012) and KITTI (Geiger et al., 2013) datasets to evaluate the model generalization of the proposed GRNet in MDE. NYUv2 is an indoor dataset. We follow the official training and testing split to evaluate our method, with 249 scenes used for training and 654 images from 215 scenes for testing. KITTI is an outdoor dataset. The images are around $376 \times 1241$ resolution. We follow the experimental setting in (Yuan et al., 2022), which consists of 85,898 training images, 1000 validation images and 500 test images. We report the results on the validation data.

### 4.2.2 Evaluation Metrics

Similar to previous work (Yuan et al., 2022; Shao et al., 2023b), we use the standard evaluation protocols in MDE, e.g., square root of the scale-invariant logarithmic error (SILog), relative absolute error (Abs Rel), relative squared error (Sq Rel), root mean squared error (RMSE), log10 error (log10), and threshold accuracy ($\delta <$1.25). We also report the computational complexity in GFLOPs.

### 4.2.3 Comparison with state-of-the-art

Table 7: Quantitative results on NYUv2. "Abs Rel" and "RMSE" are the main ranking metrics. †, ‡ and § respectively represent the use of SIFT, SuperPoint and Canny edge as the constraints of our GRNet.

| Method | Abs Rel ↓ | RMSE ↓ | Sq Rel↓ ↓ | log10↓ | $\delta <$1.25 ↑ | GFLOPs↓ |
|---|---|---|---|---|---|---|
| DORN (Fu et al., 2018a) | 0.115 | 0.509 | / | 0.051 | 0.828 | / |
| BTS (Lee et al., 2019) | 0.110 | 0.392 | 0.066 | 0.047 | 0.885 | / |
| Adabin (Bhat et al., 2021) | 0.103 | 0.364 | / | 0.044 | 0.903 | / |
| P3Depth (Patil et al., 2022) | 0.104 | 0.356 | / | 0.043 | 0.898 | / |
| NeWCRFs (Yuan et al., 2022) | 0.095 | 0.334 | 0.045 | 0.041 | 0.922 | 43.182 |
| IEBins (Shao et al., 2023b) | **0.087** | **0.314** | **0.040** | **0.038** | 0.936 | 99.068 |
| GRNet w NeWCRFs § | 0.089 | 0.314 | 0.042 | 0.039 | 0.929 | 43.182 |
| GRNet w NeWCRFs † | 0.088 | 0.314 | 0.041 | 0.039 | 0.929 | 43.182 |
| GRNet w NeWCRFs ‡ | 0.088 | **0.314** | 0.041 | **0.038** | 0.929 | **43.182** |

**Results on indoor scenes.** We choose the recent NeWCRFs as our baseline and implement GRNet On the NYUv2 dataset for indoor scenes. We use three different geometries: SIFT, SuperPoint and Canny edge. Their results are denoted by †, ‡ and § respectively. Table 7 shows that our proposed GRNet improves the depth estimation performance in the two main metrics, "Abs Rel" error and "RMSE". Specifically, the "Abs Rel" error is reduced by 7.37% relatively from 0.095 to 0.088, and the RMSE error is reduced by 5.99% relatively from 0.334 to 0.314. We also compare our method with IEBins, which proposes the Iterative Elastic Bins on top of NeWCRFs for MDE. Table 7 shows that our GRNet with NeWCRFs achieves comparable performance to IEBins but with a 56.4% reduction in GFLOPs. This shows that our method could be used to replace the iterative elastic bins. Figure 3 visualizes the comparisons between NeWCRFs and our GRNet in MDE.

It is also interesting to note that different choices of geometries improve the performance of MDE comparably, similar to that in GFSSeg. Previous work (Krishna & Vandrotti, 2023) hypothesized that the edges provide information to smooth the output of the segmentation or depth estimation for improved performance. However, this cannot explain why the keypoints improve the performance comparable to the edge. Moreover, SuperPoint and SIFT differ from each other, while their improvements in the tasks here are also similar. We conjecture that the terms from either edges or points play a role of regularization instead of smoothing.

Table 8: Quantitative results on KITTI validation. The SILog is the main ranking metric. † represents the use of SIFT as the constraint of our GRNet.

| Method | SILog ↓ | Abs Rel ↓ | Sq Rel ↓ | RMSE ↓ | $\delta <$1.25 ↑ |
|---|---|---|---|---|---|
| DORN (Fu et al., 2018a) | 12.22 | 11.78 | 3.03 | 3.80 | 0.913 |
| BTS (Lee et al., 2019) | 10.67 | 7.51 | 1.59 | 3.37 | 0.938 |
| BA-Full (Aich et al., 2021) | 10.64 | 8.25 | 1.81 | 3.30 | 0.938 |
| NeWCRFs (Yuan et al., 2022) | 8.31 | 5.54 | 0.89 | 2.55 | 0.968 |
| GRNet w NeWCRFs † | **6.93** | **4.84** | **0.76** | **2.06** | **0.979** |
| IEBins (Shao et al., 2023b) | 7.58 | 5.10 | 0.75 | 2.37 | 0.974 |
| GRNet w IEBins † | **6.87** | **4.70** | **0.74** | **2.01** | **0.980** |

**Results on outdoor scenes.** We also report the results for the outdoor scenes in KITTI. Table 8 summarizes the comparison with previous methods. By integrating the GRNet with the recent NeWCRFs and

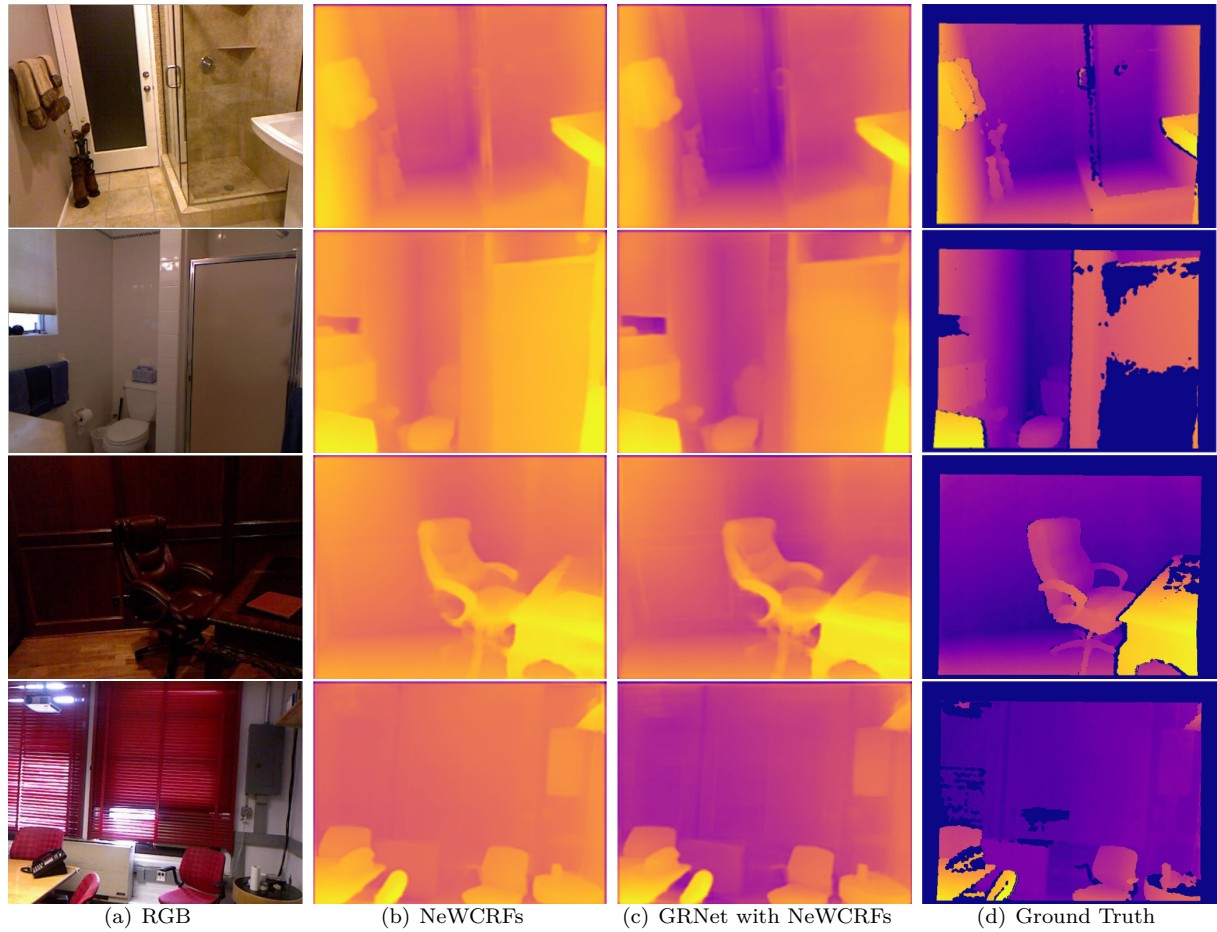

(a) RGB     (b) NeWCRFs     (c) GRNet with NeWCRFs     (d) Ground Truth

Figure 3: Some examples of monocular depth estimation from NYUv2 by NeWCRFs (Yuan et al., 2022) and our proposed GRNet

IEBins, we achieve relative reductions of the main metric SILog error by 16.6%, from 8.31 to 6.93, and 9.4%, from 7.58 to 6.87, respectively. The performance comparison on KITTI again proves that our GRNet could benefit the depth estimation methods for better generalization.

### 4.2.4 Ablation Studies

Table 9: Ablation study for NewCRFs on NYUv2 in the monocular depth estimation.

| NewCRFs | Multi-task | Primary-to-Auxiliary | Abs Rel↓ | RMSE↓ | Log10↓ | Sq Rel↓ | $\delta < 1.25$ ↑ |
|---|---|---|---|---|---|---|---|
| ✓ | | | 0.095 | 0.334 | 0.041 | 0.045 | 0.922 |
| ✓ | ✓ | | 0.091 | 0.332 | 0.040 | 0.043 | 0.923 |
| ✓ | ✓ | ✓ | **0.088** | **0.314** | **0.038** | **0.041** | **0.929** |

We perform ablation studies to evaluate the effectiveness of each proposed module. We use NeWCRFs as a baseline with the SIFT constraint. Table 9 summarizes the performance comparisons on NYUv2. Specifically, the addition of the "Multi-task" component decreases the Abs Rel metric from 0.095 to 0.091. Furthermore, integrating the primary-to-auxiliary module further decreases the Abs Rel to 0.088. The results again verify the effectiveness of our proposed GRNet.

We also conduct comparison experiments at the same training cost to evaluate the effectiveness of our algorithm. We use NeWCRFs' performance on the NYUv2 dataset as the benchmark, with a training cost of 4 V100 GPUs and a training time of 20 hours. Our proposed GRNet reduces the Abs Rel metric of NeWCRFs from 0.095 to 0.089 with the same training cost.

## 5 Conclusions

In this paper, we propose a novel regularization term to improve the training of existing deep neural networks, especially for dense prediction tasks with limited training data, such as generalized few-shot semantic segmentation and monocular depth estimation. The proposed self-trained plug-and-play geometry regularization is discarded in the inference stage and would neither modify the network structure nor change the speed. Experimental results on generalized few-shot semantic segmentation and monocular depth estimation methods validate the proposed regularisation term's effectiveness and generalisation. One limitation of the approach is the additional training time required.

### Acknowledgments

This work was supported by the Agency for Science, Technology and Research (A*STAR) under its MTC Programmatic Funds (Grant No.M23L7b0021).

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
