# OpenReview forum: "When low-vision task meets dense prediction tasks with less data: an auxiliary self-trained geometry regularization"
_TMLR — Accepted by TMLR_

### Review · Reviewer_FcoV · 2024-08-26

**Summary Of Contributions:**

This work presents a self-trained geometry regularization architecture of neural network for dense prediction tasks to improve performance for data scarce scenario.
The proposed approach and experiments seem interesting; so as long as the main concerns about structure/writing of the paper are addressed, I think this work is solid.

**Audience:**

Yes

**Claims And Evidence:**

Yes

**Requested Changes:**

Requested change:
1. As mentioned in weakness, the introduction section is a bit hard to parse.  Please restructure it so that it flows well (e.g., main problem → existing solutions → their weakness → proposed solutions → summary of results)

2. Could add short descriptions on the paragraphs in Related works 2.1 and 2.2 to further categorize the existing work

3. IEbBins → IEBins?  Please fix other typos

4.  page 6 map b and map p → outputs b and p?

5. page 6 the sentence ‘We do not change the network structure of existing methods” is a bit unclear; does it mean that the authors use the same encoder/decoder architecture to those which are typically used in some of the existing work?

6. page 8 “...to increase the task via multi-tasking.” what does it mean?

Also please address the aforementioned weakness.

**Strengths And Weaknesses:**

Strength:
The contributions themselves are clear and simple yet powerful enough to show interesting results.

Weakness:

Major

1. Introduction could be clarified more; sentences for existing work and for the proposed work are somewhat entangled and it does not flow well.

2. I feel that this paper proposes NN architecture or loss formulation rather than “Method”.  And the section 3 could be restructured to make it clearer.  If this is more than the architecture, you could write optimization problem with regularization term first in the section as the goal of this approach (including loss function).  For that you could put Figure 1 and its explanations simply in shorter paragraphs; you might add remarks or so for explanations on aggregation etc.  And then, you could discuss the choice of geometry info etc.  The current way of writing this section makes it hard to capture the overall approach in a systematic manner.

3. For experiments; the authors claim that the proposed regularization approach is suited to “low data regime”, but also show its efficiency over baseline in general.  The claim has been a bit unclear; is it only for low-data regime or is it meant to be outperforming the baseline in general?
Would like to see much clearer connection between the claim and the experimental results.
Or what happens when the number of data increase?
The flow of experiment section 4.1.3 is also a bit strange (especially the last paragraph).

4. Although mentioned in limitation, the proposed approach increases training time while it does not change computation for inference.  It would be better to mention it briefly in introduction or in related work too.

Minor

page 2 “often help the model training and reduces…” → “often helps the model training and reduces”;  please recheck other possible typos.

---

> ### Author Response · Authors · 2024-09-20
> **The response to Reviewer FcoV**
>
> We thank the reviewer for recognizing that our contributions are clear and simple yet powerful enough.
>
> Q1: As mentioned in the weakness, please restructure it so that it flows well.
>
> Reply: Thank you for the comments; we have reconstructed the introduction in the revised version, following this flow (e.g.,main problem→existing solutions→their weakness→proposed solutions→summary of results)
>
> Q2: I feel that this paper proposes NN architecture or loss formulation rather than “Method”. And the section 3 could be restructured to make it clearer. If this is more than the architecture, you could write optimization problem with regularization term first in the section as the goal of this approach (including loss function). For that you could put Figure 1 and its explanations simply in shorter paragraphs; you might add remarks or so for explanations on aggregation etc. And then, you could discuss the choice of geometry info etc. The current way of writing this section makes it hard to capture the overall approach in a systematic manner.
>
> Reply: We thank the reviewer for the comments and agree that we are proposing a new regularization term instead of a method for a specific task. We have reconstructed Section 3 in the revised version. Following the reviewer's suggestion, we first describe our proposed GRNet and its optimization objective. Then, we elaborate on the design of the geometry regularization module in detail. We also discuss the choice of low-level geometry information.
>
> Q3:  For experiments; the authors claim that the proposed regularization approach is suited to “low data regime”, but also show its efficiency over baseline in general. The claim has been a bit unclear; is it only for low-data regime or is it meant to be outperforming the baseline in general? Would like to see much clearer connection between the claim and the experimental results. Or what happens when the number of data increase? The flow of experiment section 4.1.3 is also a bit strange (especially the last paragraph).
>
> Reply: GFFSeg task is a typical low-data regime task. In the MDE task, the latest approach, DepthAnything, uses 62 million unlabeled images to pretrain a model, which is further finetuned using a smaller dataset, such as NYUv2 or KITTI, consisting of around tens of thousands of images. As the data for model finetuning is much smaller than the data used for pretraining, we consider this as low data as well. We have included these in the 'Related Work' of the revised version.
>
> Section 4.1.3 demonstrates that our proposed GRNet remains effective across different baseline algorithms, datasets, and few-shot settings. Tables 1 and 2 first detail the performance comparisons of 1-shot, 5-shot, and 10-shot on PASCAL and COCO datasets. Then, we discuss the deeper insights revealed by the experimental results in the GFSS task. As the training data for novel classes increases, the gain provided by GRNet reduces. Specifically, in the GFSS task, GRNet with POP improves the mIoU across all classes by 1.24%, 0.61%, and 0.38% in the 1-shot, 5-shot, and 10-shot settings, respectively. An increase in training data reduces the effectiveness of our proposed GRNet. However, this also highlights the ability of our method to reduce overfitting and improve generalization in low-data tasks.
>
> Q4: Although mentioned in limitation, it would be better to mention it briefly in introduction or in related work too.
>
> Reply: Thank you for the suggestions; we briefly mentioned it in the introduction and related work.
>
> Q5: Could add short descriptions on the paragraphs in Related works 2.1 and 2.2 to further categorize the existing work?
>
> Reply: We have added some descriptions to further categorize the existing work. The generalized few-shot semantic segmentation (GFSSeg) and monocular depth estimation (MDE) are first elaborated as two examples to validate our approach for dense prediction tasks with less data. Then, we also introduce the commonly used auxiliary semantic and edge information as regularization to improve the performances.
>
> Q6: please fix typos.
>
>  Reply: Thanks for your reminder; we have fixed these typos.
>
> Q7: map b and map p → outputs b and p?
>
> Reply: We have modified it to the "feature b and output p".
>
> Q8: Clarification for the sentence ‘We do not change the network structure of existing methods”.
>
> Reply: We have clarified this in the revised paper. We change it to “It seamlessly integrates the existing methods in few-shot segmentation and monocular depth estimation, without changing the network structure of existing methods. As a result, this plug-and-play module is discarded after training, incurring no additional computational cost during inference” on page 6.
>
> Q9: page 8 “...to increase the task via multi-tasking.” what does it mean?
>
> Reply: We have corrected it to “Our core idea is to introduce an extra branch as the low-level geometry regularization. This branch is included via multi-task learning.

---

> > ### Comment · Reviewer_FcoV · 2024-09-23
> > **Thank you for the response**
> >
> > Thank you for the response; I will check the updated draft, please wait a while.

---

> > > ### Comment · Reviewer_FcoV · 2024-09-24
> > > **Further clarification**
> > >
> > > Again, thank you for your responses.
> > > I have read some of the updated parts.
> > >
> > > 1. Introduction has been clarified; however, it still seems a bit entangled.
> > > For example, the sentence "Previous researchers have used edge as a supervisor signal to improve..." may need to come before you actually presenting your work in the paragraph "This work...".
> > >
> > > 2. "It is important to emphasize that our method is conceptually distinct..."  This paragraph seems too suddenly detailed.  May need to be improved.
> > >
> > > 3. Do you still think that the title of Section 3 "Method" is appropriate?  If so it is fine, but just to recheck.
> > >
> > > Please recheck the entire flow and the "claim-evidence" relation, and I don't have further concern then.

---

> > > > ### Author Response · Authors · 2024-09-24
> > > > **Response to the futher clarification**
> > > >
> > > > Again, we thank the reviewer for constructive comments.
> > > >
> > > > Q1: Introduction has been clarified; however, it still seems a bit entangled. For example, the sentence "Previous researchers have used edge as a supervisor signal to improve..." may need to come before you actually presenting your work in the paragraph "This work...".
> > > >
> > > > Reply: Thanks for your suggestion. We have reorganized the introduction and merged the sentence "Previous researchers have used edge as a supervisor signal to improve..." into the paragraph on existing solutions.
> > > >
> > > > Q2: "It is important to emphasize that our method is conceptually distinct..." This paragraph seems too suddenly detailed. May need to be improved.
> > > >
> > > > Reply: We have moved this paragraph regarding primary-to-auxiliary to Section 3.3, making the introduction clear.
> > > >
> > > > Q3: Do you still think that the title of Section 3 "Method" is appropriate? If so it is fine, but just to recheck.
> > > >
> > > > Reply. We have checked the title of Section 3. We have modified the "Method" to the "Geometry Regularization Network".

---

### Review · Reviewer_xuyb · 2024-08-26

**Summary Of Contributions:**

This paper proposes a plug-and-play geometry regularization for few-shot semantic segmentation and monocular depth estimation tasks with limited training data. This method is cost-free during the inference phase. Experimental results reveal the effectiveness across different models.

**Audience:**

Yes

**Broader Impact Concerns:**

No.

**Claims And Evidence:**

Yes

**Requested Changes:**

There are no significant flaws or issues

**Strengths And Weaknesses:**

Strengths:

This is a plug-and-play module that only operates during the training phase, introducing no additional cost for inference while improving few-shot results.

Weakness:

Due to the use of two decoders in this paper, it will inevitably result in a higher training cost. It is necessary to compare how the baseline method would perform under the same training cost.

---

> ### Author Response · Authors · 2024-09-20
> **The response to Reviewer xuyb**
>
> We thank the reviewer for recognizing that our proposed plug-and-play module does not introduce additional computational costs while improving performance effectively.
>
> Q1: Due to the use of two decoders in this paper, it will inevitably result in a higher training cost. It is necessary to compare how the baseline method would perform under the same training cost.
>
> Reply: We have also conducted the comparison experiments at the same training cost to verify the effectiveness of our algorithm. We select NeWCRFs' performance on the NYU v2 dataset as the benchmark, with a training cost of 4 V100 GPUs and a training time of 20 hours. The results show that our proposed GRNet reduces the Abs Rel metric of NeWCRFs from 0.095 to 0.089 at the same training cost. We have mentioned these in Section 4.2.4 of the revised version.

---

### Review · Reviewer_TTcv · 2024-09-07

**Summary Of Contributions:**

This paper introduces a self-trained geometry regularization framework designed to improve model performance in data-limited scenarios. It proposes a geometry aggregation module that consolidates information from multi-task learning. The effectiveness and generalization of the approach are demonstrated on generalized few-shot semantic segmentation and monocular depth estimation tasks

**Audience:**

Yes

**Claims And Evidence:**

Yes

**Requested Changes:**

None

**Strengths And Weaknesses:**

[Strength]
- The formulation of the network architecture is relatively straight-forward and rather simple, but the performance gain is quite visible. The conclusions of this paper could be instrumental to the research community.
- Solid and strong experimental results on dense prediction tasks.
- The overall presentation of the paper is good, with good descriptions of the approach.

[Weakness]
- Is this module still effective when the original model already has strong performance?  In Table 3, CAPL+GRNet shows a significant improvement in the 10-shot scenario, but the improvement for POP+GRNet is not as noticeable.
- Can you provide an ablation study for GRNet in MDE?
- Why do you train the low-level Geometry Extractor (Fig.1 (a) )?  Since you rely on low-level extractors like SIFT, SuperPoint, and Canny edge detectors to generate pseudo labels, shouldn’t the Geometry Extractor remain fixed?

---

> ### Author Response · Authors · 2024-09-20
> **The response to Reviewer TTcv**
>
> We appreciate the reviewer's positive comments on our network architecture and its potential impact on the research community. We are also grateful for the recognition of the solid experimental results and the clarity of our presentation.
>
> Q1: A concern is whether the proposed module remains effective when the original model performs well. In Table 3, while CAPL+GRNet demonstrates a significant improvement in the 10-shot scenario, the improvement for POP+GRNet is not as noticeable.
>
> Reply: Our current experimental results show that the improvement from the proposed GRNet gradually decreases as the original model improves. Specifically, in the GFSSeg task, GRNet with POP improves the mIoU across all classes by 1.24%, 0.61%, and 0.38% in the 1-shot, 5-shot, and 10-shot settings, respectively. As the training data for novel classes increases, the gain from GRNet reduces. We observe a similar trend in the monocular depth estimation (MDE) task. We conduct additional experiments to examine the effect of GRNet on NeWCRFs when the NYUv2 dataset is reduced by half. The results show that with only half of the NYUv2 training data, GRNet reduces the Abs Rel metric of NeWCRFs by 11.4%, from 0.105 to 0.093. In comparison, when using the full NYUv2 dataset, GRNet reduces the Abs Rel by 7.4%, from 0.095 to 0.088.
>
> Our method demonstrates a larger improvement for CAPL compared to POP due to different training mechanisms between the two methods. CAPL iteratively optimizes the models for the base classes and updates the novel classes in each epoch. In contrast, POP trains the models for base classes first and then updates them for the novel classes without utilizing the data of base classes. POP only uses data from novel classes to fine-tune the model. This training method limits the effectiveness of our proposed self-trained geometry regularization module in fully exploiting the available data.
>
> Q2: Can you provide an ablation study for GRNet in MDE?
>
> Reply: We have conducted an ablation study for GRNet in MDE, using NewCRF as our baseline. We evaluate the comparison experiments on the NYUv2 dataset. The results are shown in the table below. We have also incorporated the results into section 4.2.4 in the revised paper.
> | Baseline (NeWCRFs) | Multi-task | Primary-to-Auxiliary | Abs Rel | RMSE  | Log10 | Sq Rel | δ < 1.25 |
> |-------------------|------------|----------------------|---------|-------|-------|--------|----------|
> | ✔                 |            |                      | 0.095   | 0.334 | 0.041 | 0.045  | 0.922    |
> | ✔                 | ✔          |                      | 0.091   | 0.332 | 0.040 | 0.043  | 0.923    |
> | ✔                 | ✔          | ✔                    | 0.088   | 0.314 | 0.038 | 0.041  | 0.929    |

---

### Decision · Action_Editor_YzzY · 2024-11-07

**Recommendation:** Accept as is

**Comment:**

This paper has three reviewers:  The reviewers have raised various points, including the novelty and significance of the paper's contributions, the clarity of the introduction and structure of the paper, and the effectiveness of the proposed method in low-data scenarios. One reviewer leaned towards rejecting the paper, citing insufficiently significant improvements and high inference FLOPs as their main concerns. In contrast, the other two Reviewers were more positive, appreciating the paper's solid experimental results and the potential impact on the research community. They suggested that the authors address specific concerns, such as the effectiveness of the proposed module with high-performing original models, the need for an ablation study in MDE, and the clarity of the paper's structure and claims. The authors responded to these comments, providing explanations and additional experiments to support their claims. They also made revisions to the paper to address the reviewers' suggestions, such as restructuring the introduction and clarifying the methodology.  Overall, this is a good paper that shall be accepted to the journal. The authors should also make some necessary changes as suggested.

**Audience:**

This paper is primarily researchers and practitioners in the field of computer vision and deep learning.

**Claims And Evidence:**

The claims made in the paper are generally supported by evidence, but there are some concerns raised by the reviewers that suggest room for improvement in terms of clarity and convincingness: More Supporting Evidence,  Effectiveness in Low-Data Regime, and Training Cost.